# Multivariate Time-Series Forecasting with Temporal Polynomial Graph Neural Networks

**Yijing Liu**[1],    **Qinxian Liu**[1],    **Jian-Wei Zhang**[1]
**Haozhe Feng**[1],    **Zhongwei Wang**[1],    **Zihan Zhou**[1],    **Wei Chen**[1*]
[1] State Key Lab of CAD&CG, Zhejiang University, Hangzhou, China
{3150105531,22021050,zjw.cs,fenghz,wzw09,12121109,chenvis}@zju.edu.cn

## Abstract

Modeling multivariate time series (MTS) is critical in modern intelligent systems. The accurate forecast of MTS data is still challenging due to the complicated latent variable correlation. Recent works apply the Graph Neural Networks (GNNs) to the task, with the basic idea of representing the correlation as a static graph. However, predicting with a static graph causes significant bias because the correlation is time-varying in the real-world MTS data. Besides, there is no gap analysis between the actual correlation and the learned one in their works to validate the effectiveness. This paper proposes a temporal polynomial graph neural network (TPGNN) for accurate MTS forecasting, which represents the dynamic variable correlation as a temporal matrix polynomial in two steps. First, we capture the overall correlation with a static matrix basis. Then, we use a set of time-varying coefficients and the matrix basis to construct a matrix polynomial for each time step. The constructed result empirically captures the precise dynamic correlation of six synthetic MTS datasets generated by a non-repeating random walk model. Moreover, the theoretical analysis shows that TPGNN can achieve perfect approximation under a commutative condition. We conduct extensive experiments on two traffic datasets with prior structure and four benchmark datasets. The results indicate that TPGNN achieves the state-of-the-art on both short-term and long-term MTS forecastings. [1]

## 1   Introduction

The wide deployment of sensors in modern societies records tremendous time-series data, which boosts the application of energy dispatch, traffic control, etc. Commonly, there are numerous distributed sensors in a monitoring system, e.g., the temperature control system of a computing center. Univariate time-series data recorded by these sensors formulate the multivariate time-series (MTS) data. Forecasting over MTS data has been widely studied [4, 21, 41] in recent years, as it provides essential information for strategy formulation and resource schedule [30, 52].

One basic premise of modeling the MTS data is that the variables interact with each other, which is fulfilled in most cases. Therefore, capturing the variable correlation is essential for MTS forecasting. Previous deep-learning-based works like FC-LSTM [36] and TPA-LSTM [34] model the correlation with an implicit recurrent process, which is inefficient in prediction and hard to optimize. The development of graph neural networks (GNNs) [17] brings an innovative way to capture the variable dependence. Recent works [12, 25, 44] regard the MTS data as a series of graph signals and process them with GNNs, where the nodes are variables and the edge weight quantifies the dependence. Although GNNs-based methods achieve promising efficiency and accuracy in MTS forecasting,

---

[*]Corresponding author.
[1]Code is available at `https://github.com/zyplanet/TPGNN`.

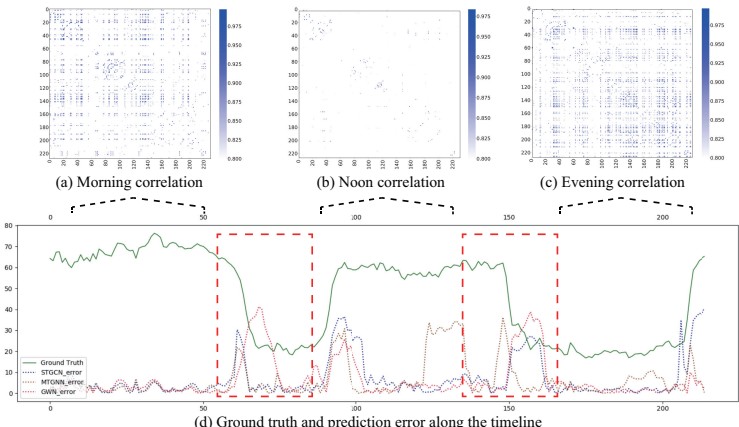

Figure 1: The sensor correlation and model prediction error in 24 hours. In (a)-(c), we apply the Pearson correlation coefficient to derive the sensor correlation of different periods, the results show time-varying patterns. We present the prediction error of three static-graph-based models in (d), the errors have steep increases between two periods, we mark them with red rectangles.

there are two notable challenges to the current approaches. 1) The MTS data have no prior variable dependence, which is required for determining the edge weight in graph construction. The existent methods compute the weight according to the physical distance among sensors [23, 35, 44] or utilize parameter matrices to learn the weight from MTS data [27, 41, 42]. As a result, the edge weights are constant in their methods. However, according to the observations from the real-world dataset, the variable dependence changes with time. Therefore, using static-weight graphs in prediction causes significant bias. Figure 1 illustrates the first challenge with a traffic dataset (PEMS-D7). It demonstrates that the correlation has distinct patterns in different periods. Besides, three static-graph-based methods [41, 42, 44] show error increases between two consecutive periods, which is caused by dependence variation. 2) Many researchers point out a gap between the actual variable dependence and the constructed correlation [25, 42, 49], i.e., the edge weight does not accurately describe the variable correlation. Although they propose various methods to bridge the gap, there is no theoretical/empirical analysis of the gap. Therefore, the method's effectiveness is still unknown.

To address the above two challenges, we propose Temporal Polynomial Graph Neural Network (TPGNN), a novel GNNs-based forecasting model for MTS data. TPGNN has an encoder-decoder structure, we encode historical MTS data with a novel GNN module and make predictions by decoding the encoded results auto-regressively. The core of TPGNN is a temporal polynomial graph (TPG) module that learns the dynamic variable dependence end-to-end. The main idea is to represent the correlation as a time-varying matrix polynomial. Firstly, we define an adjacency matrix basis $\mathbf{A}$ to capture the overall dependence for each variable pair. Then, we use a matrix polynomial of $\mathbf{A}$ to represent the correlation of each time step, where the coefficients are determined by time. To generate coefficients for MTS data of arbitrary length, we propose to use a set of cyclic timestamp embedding to index the time step. As a result, TPGNN captures the dynamic correlation with a temporal matrix polynomial, which solves the first challenge effectively. For the second challenge, we analyze the approximation error of the TPG module and provide a theoretical bound on the error. Furthermore, we evaluate the existent dependence learning methods on six synthetic MTS datasets with ground-truth variable dependence. The main contributions of the work are as follows:

- We propose a novel dependence learning module to capture MTS data's dynamic correlation, which represents the variable dependence as a temporal matrix polynomial. The resultant model achieves state-of-the-art forecasting performance on 5 of 6 benchmark datasets with significant improvements.

- We investigate the dependence approximation gap on six synthetic datasets with different levels of dependence complexity. On average, the proposed TPGNN outperforms the best baseline by $23.41\%$ on the approximation error. A theoretical result further demonstrates the dependence learning ability of TPGNN.

## 2 Related Works

**MTS Forecasting** There are two main categories of methods for MTS forecasting: implicit-dependence approaches and structural-dependence approaches. One representative method of the first category is LSTNet [20], it captures the variable dependence with convolution over variables. The structural-dependence approaches represent the variable correlation as explicit graphs and predict with GNNs. Graph Wavenet [42] (GWN) attempts to learn the variable correlation with static node representations and captures the temporal pattern with convolution neural networks (CNNs). MTGNN [41] follows the idea of GWN to learn variable correlation statically and proposes a new graph convolution module for MTS forecasting. Although works like GMAN [50], SLC [49], and StemGNN [4] model a dynamic correlation among variables with self-attention, the learned graph is sensitive to the input, causing a significant forecasting variance. Our construction is based on a set of time-variant coefficients, the result is independent of the input and robust in forecasting. Some works that focus on learning time-varying graph structure [26, 33, 45]. However, these graph structures are not captured from the forecast task directly. They thus cause bias and increase the computation cost. There are some transformer-based works [43, 46, 48], but their methods do not thoroughly discuss the dynamic-dependence issues.

**GNNs and Polynomial Graph Filters (PGFs)** GNNs are based on the message-passing mechanism. One representative work is GraphSAGE [14], which learns a sample-aggregation function to generate the node embedding. Michael et al. [6] generalize the convolutional neural networks (CNNs) to the graph structure. Many filters of GNNs belong to PGFs, and Gama et al. [11] discussed the connections. In order to incorporate the node feature of high-order neighbors, many works introduce PGFs to their methods. APPNP [18] and SGC [40] adopt fixed-weight PGFs and use hyperparameters to define the coefficients. However, the constructed fixed-weight PGFs are biased because the coefficients are task-irrelevant. GPR-GNN [5] constructs a PGF from the graph dataset, where the coefficients are learnable. Besides, there are also some PGFs-based methods for MTS data learning [13, 15, 16]. Although these works are competitive in capturing graph structure, they fail in learning the dynamic correlation, we illustrate the point in Section 5.4. Moreover, the missing/noisy/incomplete adjacency matrix causes difficulties in applying these methods. Our approach introduces a self-adaptive graph to the PGFs construction, which improves the dependence capturing ability and the model versatility.

## 3 The Framework of TPGNN

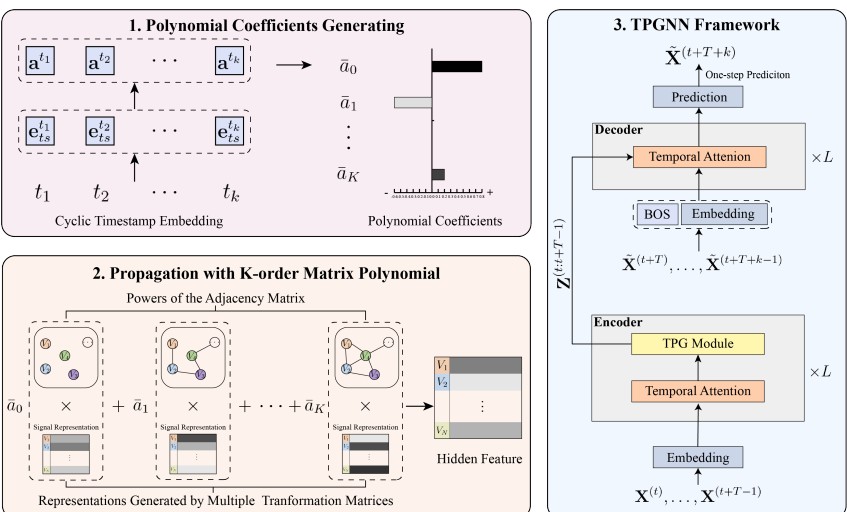

Figure 2: Overview of the proposed TPGNN. (1) The TPG module generates a set of coefficients according to the time steps. (2) We define a matrix polynomial with the coefficients to capture the dependence. (3) TPGNN has an encoder-decoder structure and predicts in an auto-regressive manner.

In this section, we first define the MTS forecasting task in the context of graph signal processing. We then introduce the TPG module and illustrate the inference pipeline of our framework.

### 3.1 Problem Defintion

At time step $t$, the MTS data is a signal set $\mathcal{G}^{(t)} = \{\mathcal{V}, \mathcal{E}^{(t)}, \mathbf{X}^{(t)}, \mathbf{W}^{(t)}\}$. The node set $\mathcal{V}$ ($|\mathcal{V}| = N$) contains $N$ variables of MTS data, $\mathcal{E}^{(t)}$ is the edge set, $\mathbf{X}^{(t)} \in \mathbb{R}^{N \times 1}$ records the signal of $N$ variables, and $\mathbf{W}^{(t)} \in \mathbb{R}^{N \times N}$ is a weighted adjacency matrix. If variables $i$ and $j$ are dependent, then an edge connects the two nodes, and entry $\mathbf{W}_{ij}^{(t)}$ indicates their correlation. Otherwise, $\mathbf{W}_{ij}^{(t)}$ equals to zero. Given $T$ observations of the signal sets, our problem is to find a function $F$ that can forecast the variable state of the subsequent $T'$ signals:

$$(\mathcal{G}^{(t)}, \mathcal{G}^{(t+1)}, \ldots, \mathcal{G}^{(t+T-1)}) \xrightarrow{F} \mathbf{X}^{(t+T)}, \mathbf{X}^{(t+T+1)}, \ldots, \mathbf{X}^{(t+T+T'-1)}. \tag{1}$$

The setting regards the graph structure as a latent feature of MTS data, and the forecast target is the same as the standard setting. Prevailing works [25, 41, 42] regard $\mathbf{W}^{(t)}$ as a time-invariant matrix, while in our work, $\mathbf{W}^{(t)}$ changes with the time.

### 3.2 Represent the Correlation as a Temporal Matrix Polynomial

To capture $\mathcal{E}$ and $\mathbf{W}^t$ for MTS data, we construct a series of matrix polynomials that share a matrix basis. The TPG module controls the influence of each polynomial term temporally with a set of adaptive coefficients. As a result, we can approximate a wide range of dynamic graph structures.

**Counstructing the adjacency matrix basis.** We firstly define an initial adjacency matrix $\mathbf{A} \in \mathbb{R}^{N \times N}$ following the self-adaptive graph proposed by Wu et al. [42]. They learn variable embeddings $\mathbf{E} \in \mathbb{R}^{N \times c}$ ($c$ is the embedding dimension) from MTS data and define the variable dependence with embedding similarity. Besides, they further remove weak connections with $\mathrm{ReLU}(\cdot)$ and normalize the result with $\mathrm{SoftMax}(\cdot)$.

$$\mathbf{A} = \mathrm{SoftMax}(\mathrm{ReLU}(\mathbf{E}\mathbf{E}^{\mathbf{T}})) \tag{2}$$

For MTS data that have a prior structure, e.g., the physical distance among sensors, we denote the corresponding adjacency matrix as $\mathbf{W} \in \mathbb{R}^{N \times N}$. To incorporate $\mathbf{W}$ into the encoding process, we first calculate its symmetric normalized Laplacian [17] $\mathbf{L} = \mathbf{D}^{-\frac{1}{2}}(\mathbf{I} + \mathbf{W})\mathbf{D}^{-\frac{1}{2}}$ ($\mathbf{I}$ is the identity matrix and $\mathbf{D}$ is the degree matrix), then combine the two results to define the initial adjacency matrix:

$$\mathbf{A} = \mathrm{SoftMax}(\mathrm{ReLU}(\mathbf{E}\mathbf{E}^{\mathbf{T}})) + \mathbf{L} \tag{3}$$

**Generating the temporal polynomial coefficients.** At time step $t$, we represent the variable dependence as a $K$-order matrix polynomial of $\mathbf{A}$:

$$\mathbf{W}^{(t)} = \Sigma_{k=0}^{K} a_k^{(t)} \mathbf{A}^k \tag{4}$$

Due to the polynomial coefficients being time-variant, $\mathbf{W}^{(t)}$ shows dynamic behavior. However, learning coefficients for every time step is impractical because the sequence length grows infinitely. Fortunately, the periodicity property is common in MTS data [8, 29]. Therefore, we propose to use a set of cyclic timestamp embeddings to index the time step. Firstly, we define $T_p$ timestamp embeddings $(\mathbf{e}_{ts}^{(1)}, \ldots, \mathbf{e}_{ts}^{(T_p)}) \in \mathbb{R}^{D_e}$, where $T_p$ is the cycle and $D_e$ is the embedding dimension. Following a cyclic order, we then assign the $(t\%T_p)$-th embedding to time step $t$, where $\%$ is the remainder operator. Finally, we generate the coefficients for time step $t$ by the $\mathbf{e}_{ts}^{(t\%T_p)}$. As shown in Figure 2 (1), Given $T$ historical graph signals start from time step $t$, we calculate the coefficients $\mathbf{a} \in \mathbb{R}^{K+1}$ for each graph signal with a coefficient matrix $\mathbf{W}_c \in \mathbb{R}^{D_e \times (K+1)}$.

$$(\mathbf{a}^{(t)}, \ldots, \mathbf{a}^{(t+T-1)}) = (\mathbf{e}_{ts}^{(t\%T_p)}, \ldots, \mathbf{e}_{ts}^{(t+T-1\%T_p)})\mathbf{W}_c \tag{5}$$

According to Equation 4, each $\mathbf{a}^{(t)}$ defines a $K$-order matrix polynomial of $\mathbf{A}$, representing variable dependence of time step $t$. Instead of predicting with $T$ different matrix polynomials, we use an average result to improve the prediction efficiency and robustness. Due to the linearity of the polynomial, the average polynomial is equivalent to the polynomial with average coefficients. We calculate the final coefficients $\bar{\mathbf{a}}$ with another parameter matrix $\mathbf{W}_a \in \mathbb{R}^{T \times 1}$ (average matrix).

$$\bar{\mathbf{a}} = (\bar{a}_0, \bar{a}_1, \ldots, \bar{a}_K) = (\mathbf{a}^{(t)}, \ldots, \mathbf{a}^{(t+T-1)})\mathbf{W}_a \tag{6}$$

In Figure 2 (2), we illusrate the TPG module's propagation process with $\bar{\mathbf{a}}$. Let $\mathbf{X} \in \mathbb{R}^{N \times D_e}$ denote the matrix signal embedding, $\mathbf{W}_k \in \mathbb{R}^{D_e \times D_e}, k = 0, \ldots, K$ denote the model parameter matrices, $|| \cdot ||_F$ denote the Frobenius norm of matrix, we derive the hidden feature $\mathbf{Z}^{(t)} \in \mathbb{R}^{N \times D_e}$ as follows:

$$\mathbf{Z}^{(t)} = \Sigma_{k=0}^{K} \bar{a}_k \mathbf{A}^k \mathbf{X}^{(t)} \frac{\mathbf{W}_k}{||\mathbf{W}_k||_F} \tag{7}$$

TPG module follows the diffusion graph convolution layer [23] to introduce $(K + 1)$ parameter matrices for enhancing representation diversity. Besides, we decouple the parameter matrix norm and coefficient by normalizing $\mathbf{W}_k$ with a factor of $\frac{1}{||\mathbf{W}_k||_F}$, since $\mathbf{W}_k$ with a large norm increases the $k$-th term's contribution. As a result, $\bar{\mathbf{a}}$ completely controls the contribution of each term.

### 3.3 Inference Pipeline

In Figure 2 (3), TPGNN has an encoder-decoder structure. First, we derive the historical data embedding with a linear transformation, then feed the results into the encoder. The encoder generates data encoding $\mathbf{Z}^{(t:t+T-1)} \in \mathbb{R}^{T \times N \times D_e}$ for time step $t$ to $t + T - 1$ with the TPG module and temporal attention layer. The temporal attention layer follows the design of Transformer [37], which uses self-attention to capture intra-series patterns for each variable. Finally, the decoder forecasts with the data encoding and temporal attention layer. For the first prediction of time step $t + T$, the decoder queries the data encoding with a learnable beginning of sentence (BOS) token $\mathbf{E}_{BOS} \in \mathbb{R}^{N \times D_e}$. The query result represents the first prediction, denoted as $\mathbf{E}_X^{(t+T)} \in \mathbb{R}^{N \times D_e}$. We use a prediction matrix $\mathbf{W}_{pred} \in \mathbb{R}^{D_e \times 1}$ to get the forecasting result $\tilde{\mathbf{X}}^{(t+T)} \in \mathbb{R}^{N \times 1}$.

$$\mathbf{E}_X^{(t+T)} = \text{Decoder}(\mathbf{E_{BOS}}, \mathbf{Z}^{(t:t+T-1)}),$$
$$\tilde{\mathbf{X}}^{(t+T)} = \mathbf{E}_X^{(t+T)} \mathbf{W}_{pred}, \tag{8}$$

After generating the subsequent $k$ query results $(\mathbf{E}_X^{(t+T)}, \ldots, \mathbf{E}_X^{(t+T+k-1)})$, the decoder derives the $(k+1)$-th result $\mathbf{E}_X^{(t+T+k)}$ by querying $\mathbf{Z}^{(t:t+T-1)}$ with the latest $L_{max}$ query results. The forecast basically follows an auto-regressive (AR) mechanism, where $L_{max}$ is the maximum query length.

$$\mathbf{E}_X^{(t+T+k)} = \text{Decoder}((\mathbf{E}_X^{(t+T+k-L_{max})}, \ldots, \mathbf{E}_X^{(t+T+k-1)}), \mathbf{Z}),$$
$$\tilde{\mathbf{X}}^{(t+T+k)} = \mathbf{E}_X^{(t+T+k)} \mathbf{W}_{pred}. \tag{9}$$

The above process continues until getting the subsequent $T'$ forecasts, such AR mechanism helps TPGNN to capture long-term intra-series dependence with informative context.

## 4 Theoretical Properties of TPGNN

To explore the dependence learning capability of TPGNN, we focus on analyzing the theoretical gap between the optimal graph structure and the TPG module's approximation in this section.

**TPGNN is able to achieve perfect approximation.** Unlike the real function, the approximation of graph structure largely depends on the topology property of the target graph, where the lower bound of the approximation error is nonzero in many cases. However, TPGNN mitigates the gap for a wide range of dynamic graph structures with a small group of parameters, which leads to a robust and accurate forecasting result. The approximation capability of TPGNN is concluded as the following result, we show the detailed statement and proofs in Appendix A.9 due to the space limitation.

**Theorem 1**. Let $\mathbf{G}^{(1)}, \ldots, \mathbf{G}^{(T)} \in \mathbb{R}^{N \times N}$ be the symmetric normalized Laplacian of the optimal graph structure for time step 1 to $T$, $\mathbf{A} \in \mathbb{R}^{N \times N}$ is the initial adjacency matrix of TPGNN. We model these Laplacians with Equation 4, and the corresponding approximation error $e^{(1:T)} = \frac{1}{T} \Sigma_{t=1}^{T} ||\mathbf{W}^{(t)} - \mathbf{G}^{(t)}||_F^2$, where the $|| \cdot ||_F$ is the Frobenius norm of the matrix. Suppose all the matrices are symmetric (undirected graph), $\mathbf{G}_i \mathbf{G}_j = \mathbf{G}_j \mathbf{G}_i, \forall i, j$, $\mathbf{A}$ has $N$ different singular values, and the polynomial's order is large enough. Then we have the following estimation for $e^{(1:T)}$.

$$(1 - \lambda_{max}) \mathbb{E}_t ||\mathbf{G}^{(t)}||_F^2 \leq e^{(1:T)} \leq (1 - \lambda_{min}) \mathbb{E}_t ||\mathbf{G}^{(t)}||_F^2, \tag{10}$$

where $\mathbb{E}_t||\mathbf{G}^{(t)}||_F^2 = \frac{1}{T}\Sigma_{t=1}^T||\mathbf{G}^{(t)}||_F^2$ is the average norm of the Laplacians, $\lambda_{min}, \lambda_{max} \in [0, 1]$ are the minimum and maximum eigenvalues of a constant matrix. If $\mathbf{A}$ is commutative with $\mathbf{G}^{(t)}$, then $\lambda_{max} = \lambda_{min} = 1$, i.e., we achieve a uniform perfect approximation. Although the zero error depends on the commutative condition, $\mathbf{A}$ is learnable with Equation 2, which relaxes the restriction. Moreover, TPGNN empirically achieves the best approximation compared with other methods.

## 5 Experiments

Table 1: Dataset statistics.

| Datasets | # Samples | # Nodes | Sample Rate | Input Length | Output Length |
|---|---|---|---|---|---|
| Traffic | 17,544 | 862 | 1 hour | 168 | 1 |
| Solar-Energy | 52,560 | 137 | 10 minutes | 168 | 1 |
| Electricity | 26304 | 321 | 1 hour | 168 | 1 |
| Exchange-Rate | 7,588 | 8 | 1 day | 168 | 1 |
| PEMS-D7 | 12672 | 228 | 5 minutes | 12 | 12 |
| PEMS-Bay | 52116 | 325 | 5 minutes | 12 | 12 |

### 5.1 Experimental Setup

We validate the performance of TPGNN on two tasks: single-step and multi-step MTS forecasting. In Table 1, we present the statistics of six benchmark datasets. The first four benchmark datasets [2, 19] aim to predict a single future step; the last two datasets [23,44] record a distance matrix among sensors, where the goal is to forecast multiple future steps. More details about the datasets are given in the AppendixA.3. For single-step prediction, the baselines are VAR-MLP [47], GP [31], RNN-GRU [41], LSTNet [20], TPA-LSTM [34], and MTGNN [41]. MTGNN firstly introduces GNNs to general MTS data and achieves state-of-the-art (SOTA) performance; other baselines are representative

Table 2: We forecast 3, 6, 12, and 24 horizons for the first four real-world datasets, and models with high CORR and low RSE are preferred. TPGNN outperforms other models on three of the first four datasets. For the remaining two datasets, we predict the next 15, 30, and 60 minutes based on one-hour observation, and the three metrics represent the prediction error. The results show that TPGNN achieves state-of-the-art performance on both short-term and long-term predictions.

| Dataset | | Solar-Energy | | | | Traffi | | | | Electricity | | | | Exchange-Rate | | | |
|---|---|---|---|---|---|---|---|---|---|---|---|---|---|---|---|---|---|
| | | Horizon | | | | Horizon | | | | Horizon | | | | Horizon | | | |
| Methods | Metric | 3 | 6 | 12 | 24 | 3 | 6 | 12 | 24 | 3 | 6 | 12 | 24 | 3 | 6 | 12 | 24 |
| VARMLP | RSE | 0.1922 | 0.2679 | 0.4244 | 0.6841 | 0.5582 | 0.6579 | 0.6023 | 0.6146 | 0.1392 | 0.1620 | 0.1557 | 0.1274 | 0.0265 | 0.0394 | 0.0407 | 0.0578 |
| VARMLP | CORR | 0.9829 | 0.9655 | 0.9058 | 0.7149 | 0.8245 | 0.7695 | 0.7929 | 0.7891 | 0.8708 | 0.8389 | 0.8192 | 0.8679 | 0.8609 | 0.8725 | 0.8280 | 0.7675 |
| GP | RSE | 0.2259 | 0.3286 | 0.5200 | 0.7973 | 0.6082 | 0.6772 | 0.6406 | 0.5995 | 0.1500 | 0.1907 | 0.1621 | 0.1273 | 0.0239 | 0.0272 | 0.0394 | 0.0580 |
| GP | CORR | 0.9751 | 0.9448 | 0.8518 | 0.5971 | 0.7831 | 0.7406 | 0.7671 | 0.7909 | 0.8670 | 0.8334 | 0.8394 | 0.8818 | 0.8713 | 0.8193 | 0.8484 | 0.8278 |
| RNN-GRU | RSE | 0.1932 | 0.2628 | 0.4163 | 0.4852 | 0.5358 | 0.5522 | 0.5562 | 0.5633 | 0.1102 | 0.1144 | 0.1183 | 0.1295 | 0.0192 | 0.0264 | 0.0408 | 0.0626 |
| RNN-GRU | CORR | 0.9823 | 0.9675 | 0.9150 | 0.8823 | 0.8511 | 0.8405 | 0.8345 | 0.8300 | 0.8597 | 0.8623 | 0.8472 | 0.8651 | 0.9786 | **0.9712** | 0.9513 | 0.9223 |
| LSTNet | RSE | 0.1843 | 0.2559 | 0.3254 | 0.4643 | 0.4777 | 0.4893 | 0.4950 | 0.4973 | 0.0864 | 0.0931 | 0.1007 | 0.1007 | 0.0226 | 0.0280 | 0.0356 | 0.0449 |
| LSTNet | CORR | 0.9843 | 0.9690 | 0.9467 | 0.8870 | 0.8721 | 0.8690 | 0.8614 | 0.8588 | 0.9283 | 0.9135 | 0.9077 | 0.9119 | 0.9735 | 0.9658 | 0.9511 | 0.9354 |
| TPA-LSTM | RSE | 0.1803 | **0.2347** | 0.3234 | 0.4389 | 0.4487 | 0.4658 | 0.4641 | 0.4765 | 0.0823 | 0.0916 | 0.0964 | 0.1006 | 0.0174 | **0.0241** | **0.0341** | **0.0444** |
| TPA-LSTM | CORR | 0.9850 | **0.9742** | 0.9487 | 0.9081 | 0.8812 | 0.8717 | 0.8717 | 0.8629 | 0.9439 | 0.9337 | 0.9250 | 0.9133 | 0.9790 | 0.9709 | **0.9564** | **0.9381** |
| MTGNN | RSE | **0.1778** | 0.2348 | 0.3109 | 0.4270 | 0.4162 | 0.4754 | **0.4461** | **0.4535** | 0.0745 | 0.0878 | 0.0916 | 0.0953 | 0.0194 | 0.0259 | 0.0349 | 0.0456 |
| MTGNN | CORR | **0.9852** | 0.9726 | 0.9509 | 0.9031 | 0.8963 | 0.8667 | 0.8794 | 0.8810 | **0.9474** | 0.9316 | 0.9278 | 0.9234 | 0.9786 | 0.9708 | 0.9551 | 0.9372 |
| TPGNN | RSE | 0.1850 | 0.2412 | **0.3059** | **0.3498** | **0.3989** | **0.4715** | 0.4476 | 0.4696 | **0.0627** | **0.0685** | **0.0699** | **0.0936** | **0.0174** | 0.0250 | 0.0350 | 0.0458 |
| TPGNN | CORR | 0.9840 | 0.9716 | **0.9529** | **0.9710** | **0.9232** | **0.8945** | **0.9028** | **0.8858** | 0.9417 | **0.9362** | **0.9285** | **0.9293** | **0.9792** | 0.9687 | 0.9509 | 0.9306 |

| Model | PEMS-BAY (Horizon 3/6/12) | | | PEMS-D7 (Horizon 3/6/12) | | |
|---|---|---|---|---|---|---|
| | MAE | MAPE(%) | RMSE | MAE | MAPE(%) | RMSE |
| ARIMA | 1.62/2.33/3.38 | 3.50/5.40/8.30 | 3.30/4.76/6.50 | 5.55/5.86/6.27 | 12.92/13.94/15.20 | 9.00/9.13/9.38 |
| FC-LSTM | 2.05/2.20/2.37 | 4.80/5.20/5.70 | 4.19/4.55/4.96 | 3.57/3.92/4.16 | 8.60/9.55/10.10 | 6.20/7.03/7.51 |
| STGCN | 1.39/1.84/2.42 | 3.00/4.22/5.58 | 2.92/4.12/5.33 | 2.25/3.03/4.02 | 5.26/7.33/9.85 | 4.04/5.70/7.64 |
| DCRNN | 1.38/1.74/2.07 | 2.90/3.90/4.90 | 2.95/3.97/4.74 | 2.25/2.98/3.83 | 5.30/7.39/9.85 | 4.04/5.58/7.19 |
| StemGNN | 1.52/1.94/2.45 | 3.38/4.58/6.03 | 3.06/4.07/5.04 | 2.94/3.66/4.66 | 7.63/9.66/12.58 | 5.05/6.35/8.00 |
| Graph WaveNet | 1.30/**1.63**/1.95 | 2.73/3.67/4.63 | 2.74/3.70/4.52 | 2.18/2.95/3.88 | 5.02/7.22/10.03 | 4.18/5.82/7.61 |
| Informer | 2.30/2.40/2.55 | 5.02/5.32/5.73 | 4.21/4.49/4.85 | 3.64/3.77/4.09 | 8.66/9.07/9.87 | 6.02/6.34/6.85 |
| MTGNN | 1.32/1.65/**1.94** | 2.77/3.69/4.53 | 2.79/3.74/**4.49** | 2.17/2.89/4.02 | 5.03/6.93/9.93 | **4.01**/5.84/8.78 |
| TPGNN | **1.26**/1.65/2.05 | **2.56/3.47/4.40** | **2.64/3.65**/4.58 | **2.12/2.72/3.22** | **5.00/6.73/8.22** | 4.05/**5.45/6.56** |

non-GNNs-based methods. For multi-step prediction, the baselines are ARIMA [23],FC-LSTM [36], DCRNN [22], STGCN [44], StemGNN [4], Graph WaveNet, Informer [51], and MTGNN; all the baselines are GNN-based except ARIMA, FC-LSTM and Informer. These GNNs-based baselines are representative methods for MTS data with a prior structure. Besides, the Informer is a novel Transformer-based model that achieves SOTA performance on long-term forecasting. We list the paramter scale of each model in Appendix A.4 to show our method is light-weight. We use five evaluation metrics [41]: Mean Absolute Error (MAE), Mean Absolute Percentage Error (MAPE), Root Mean Squared Error (RMSE), Root Relative Squared Error (RRSE), and Empirical Correlation Coefficient (CORR). For CORR, higher values are better. For the other metrics, lower values are better. We tune the hyperparameters on the validation dataset, the results are presented in the Appendix A.7. We divide the dataset into three parts for training, validation, and testing with a ratio of 7:1:2. More details about the baselines, metrics, and experiment settings are in Appendix A.4A.5A.6.

## 5.2 Main Results

Table 2 summarizes the experimental results of TPGNN. Generally, TPGNN achieves the state-of-the-art on most of the datasets. Our framework has an on-par performance for the single-step forecasting task with the SOTA methods like MTGNN and TPA-LSTM. Moreover, TPGNN makes a significant improvement on Traffic and Electricity datasets. On the Solar-Energy, TPGNN lowers RSE by $1.61\%, 18.08\%$ and increases CORR by $1.47\%, 6.93\%$ over the horizons of 12, 24. Our method improves the CORR by $2.21\%$ on average over the four horizons of the Traffic. Moreover, we lower the RSE by $15.82\%$ on average over the four horizons of Electricity. Our method fails to achieve SOTA performance on the exchange-rate data. We think the main reason is the small sample size, which causes difficulties in capturing the dynamic variable dependence. The second row of Table 2 concludes the results of the multi-step forecasting task, which demonstrates that TPGNN achieves SOTA performance on the two datasets with prior structure. On the PEMS-BAY, TPGNN lowers the MAPE by $5.47\%$ on average over the three horizons. Besides, The RMSE is reduced by $4.30\%, 2.41\%$ over the horizons of 3 and 6. TPGNN reduces the MAE/MAPE by $8.04\%/6.61\%$ on average over the three horizons on the PEMS-D7 dataset. The RMSE is lowered by $2.33\%, 4.23\%$ over the horizon of 6 and 12. Although the prior structure is provided to each baseline, the results on the last two traffic datasets demonstrate that TPGNN can capture more precise variable dependence and intra-series patterns than other GNNs-based methods.

## 5.3 Ablation Experiments

Table 3: Long-term prediction results for ablation study on the PEMS-D7 dataset.

| Metrics | TPGNN | w/o TPG | w/o dynamic | w/o overview | w/o normalize | w/o $K$-matrices |
|---|---|---|---|---|---|---|
| MAE | **3.215**±**0.014** | 3.701±0.021 | 3.480±0.047 | 3.330±0.031 | 3.261±0.035 | 3.294±0.036 |
| MAPE(%) | **8.221**±**0.035** | 9.553±0.106 | 8.873±0.014 | 8.539±0.118 | 8.347±0.061 | 8.406±0.245 |
| RMSE | **6.559**±**0.023** | 7.473±0.033 | 6.989±0.084 | 6.806±0.098 | 6.671±0.033 | 6.701±0.065 |

The PEMS-D7 includes a prior structure and has complex inter-series and intra-series dependence, we conduct ablation experiments on the PEMS-D7 to evaluate the effectiveness of each design in our method. Due to the importance and complexity of the long-term prediction, we set the horizon as 12. More results of other horizons can be found in Appendix A.8. There are five experiments in total, we repeat each experiment 5 times and report the average and standard deviation (std). **w/o TPG**: TPGNN without the TPG module. We replace the graph constructed by TPG with the prior structure of PEMS-D7. **w/o dynamic**: TPGNN without the time-varying coefficients. We adopt a set of learnable static coefficients to construct the graph. **w/o overview**: TPGNN does not use average coefficients in Equation 6. We predict with $T$ graphs defined by Equation 5. **w/o normalize**: TPGNN without the normalization factor in Equation 7. **w/o K-matrices**: TPGNN without multiple parameter matrices in Equation 7. We use an identical normalized matrix to derive the hidden feature. Table 3 summarizes the experimental results, which indicates that all the designs are indispensable. The result of w/o TPG illustrates the TPG module's critical contribution to our method. Although we replace the graph constructed by the TPG module with the prior structure given by PEMS-D7, the performance has a steep decline by $13.13\%, 13.94\%, 12.23\%$ on the three metrics. w/o dynamic follows the strategy of GPR-GNN [5] that uses static coefficients to capture graph structure from data. However, the results indicate the importance of introducing time-variant coefficients, which

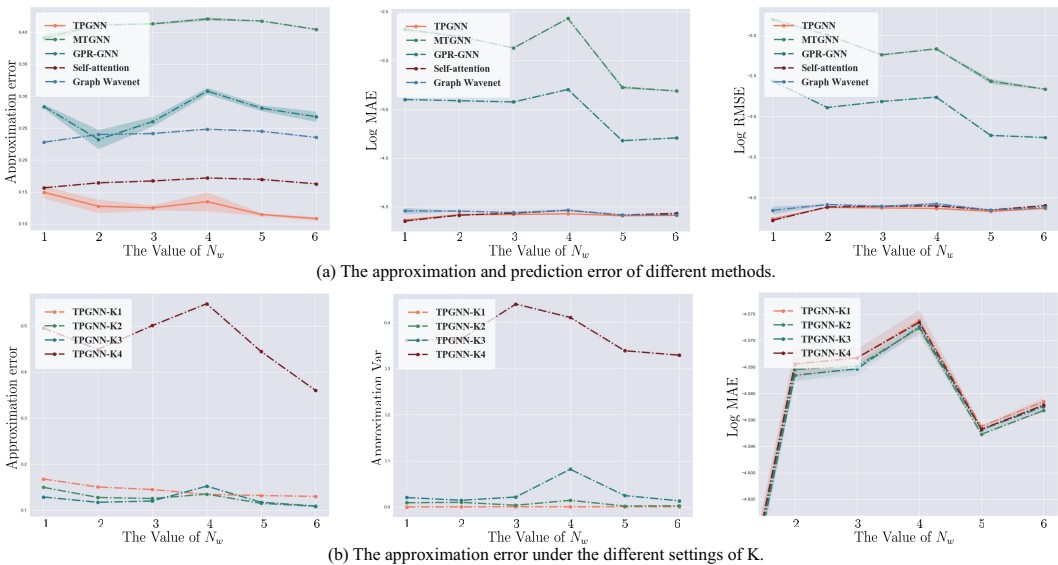

(a) The approximation and prediction error of different methods.

(b) The approximation error under the different settings of K.

Figure 3: The approximation error and prediction error of different graph approximation methods. (a) compares TPGNN with other baselines, the results indicate TPGNN significantly reduces the approximation error and achieves the highest prediction accuracy. (b) investigates the effect of polynomial order to the dependence approximation error. The high-order polynomials impairs the precision and robustness of the dependence learning.

significantly improve the long-term prediction accuracy by $7.04\%$ on average. w/o overview and w/o K-matrices show large variance compared with other results, indicating the two modules are essential for improving model robustness. Besides, the w/o overview's performance is close to the w/o dynamic, demonstrating the necessity of using the average coefficients. w/o normalize has comparable performance to the TPGNN, possibly due to the weight decay causing similar norms. Nevertheless, it demonstrates that the normalization helps TPGNN capture the variable dependence.

Table 4: Average results on six synthetic datasets configured with different $N_w$.

| Method | TPGNN | Graph Wavenet | MTGNN | GPR-GNN | Self-attention |
|---|---|---|---|---|---|
| MFE | **0.1266**±$7.057\times10^{-3}$ | 0.2397±$3.476\times10^{-4}$ | 0.4098±$1.981\times10^{-3}$ | 0.2721±$7.060\times10^{-3}$ | 0.1653±$1.469\times10^{-6}$ |
| MAE | **0.01015**±$1.442\times10^{-5}$ | 0.01049±$10.30\times10^{-5}$ | 0.05669±$40.29\times10^{-5}$ | 0.03008±$1.638\times10^{-5}$ | 0.01026±$1.289\times10^{-5}$ |
| RMSE | **0.01568**±$1.986\times10^{-5}$ | 0.01632±$2.276\times10^{-4}$ | 0.1114±$1.344\times10^{-3}$ | 0.05561±$3.537\times10^{-5}$ | 0.01593±$2.051\times10^{-5}$ |

## 5.4 Study of the Graph Structure Approximation Gap

**Synthetic data.** To investigate the empirical gap between the optimal graph structure and the approximation result, we propose to generate MTS data with ground-truth dynamic dependence using a non-repeating random walk (NRW) model [7], which is widely adopted in time-series data generation [1, 32]. At time step $t$, we synthesize the matrix signal $\mathbf{X}^{(t)} \in \mathbb{R}^{N \times 1}$ with a dynamic weighted adjacency matrix $\mathbf{W}^{(t)}$ by $\mathbf{X}^{(t)} = \mathcal{N}(\mathbf{W}^{(t-1)}\mathbf{X}^{(t-1)}, \sigma)$, where $\mathcal{N}(\cdot, \cdot)$ is the normal distribution, $\sigma \in \mathbb{R}$ controls the variance, and $\mathbf{X}^{(0)}$ is sampled from a discrete uniform distribution. We define $\mathbf{W}^{(t)}$ by traversing $N_w$ constant matrices $(\mathbf{G}^{(1)}, \ldots, \mathbf{G}^{(N_w)})$ in a cyclic order, where each $\mathbf{G}^{(i)}$ is a Laplacian of sparsified random adjacency matrix. Given a period of $T_p, T_p \geq N_w$, we partition the period into $N_w$ even intervals of length $T_s$, and each interval shares one of the matrices, i.e., $\mathbf{W}^{(t)} = \mathbf{G}^{([(t\%T_p)/T_s])}$, where $[\cdot]$ is the rounding function. We point out that $N_w$ controls the complexity of dynamic dependence, and $N_w = 1$ corresponds to a static-dependence case. To increase the data diversity, we randomly initialize the $X^{(t)}$ for every $T_p$ step. We set $T_p = 120, \sigma = 0.001$ in our evaluation and generate six MTS data of length 2400 with $N_w = 1, \ldots, 6$. Besides, each

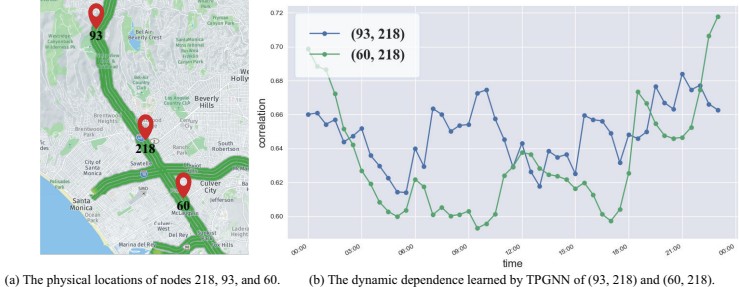

(a) The physical locations of nodes 218, 93, and 60.  (b) The dynamic dependence learned by TPGNN of (93, 218) and (60, 218).

Figure 4: Three nodes are highly correlated in the TPG's learned result but independent in the prior structure. We find they are on the same road and have different dynamic correlation trends.

synthetic dataset is divided into three parts for training, validation, and testing with a ratio of 7:1:2. More details about the synthetic algorithm, configuration, and synthetic results are in Appendix A.3.

**Predicting with different graph approximation methods.** We evaluate the dependence capturing ability of different methods on the six synthetic datasets, where the task is the next-step prediction based on current variable states. Suppose that $\tilde{\mathbf{W}}^{(t)} \in \mathbb{R}^{N \times N}, t = 0, \ldots, T-1$ is the learned adjacency matrix series, we measure the approximation gap with a Mean Frobenius Error (MFE) metric, defined by $MFE = \frac{1}{TN^2} \Sigma_{t=0}^{T-1} ||\mathbf{W}^t - \tilde{\mathbf{W}}^{(t)}||_F$. Besides, we use MAE/RMSE to measure the forecasting accuracy. The baselines are Graph Wavenet (GWN), MTGNN, GPR-GNN, and Self-attention. The first three methods are static-dependence-based. Self-attention constructs a dynamic graph, which is adopted in many works like GMAN [50] and StemGNN [4]. We follow the implementation of StemGNN to learn the graph structure with a GRU module and self-attention operator in the evaluation. The polynomial order $K$ is fixed as 2 in the evaluation, we repeat each experiment 5 times and report the average/std, more details about the settings are in Appenedix A.6. The main results are summarized in Figure 3 (a) and Table 4, demonstrating that TPGNN outperforms other baselines on the six synthetic datasets. In Table 4, TPGNN reduces the graph approximation error by $23.41\%$ on average over the six datasets. The first chart of Figure 3 (a) also illustrates the improvement, we consistently achieve the best approximation under different $N_w$. Moreover, the MFE of TPGNN has a descending trend with the increase of $N_w$ while other methods' approximation errors are increasing. We think the TPG module overfits on datasets with small $N_w$, since their behavior tends to have static dependence. TPGNN achieves the lowest prediction error on average, the improvement in the accuracy is slight compared with MFE because the task is a simple next-step prediction. Furthermore, we observe that the ranking on MFE is identical to the MAE/RMSE, which demonstrates the importance of capturing precise variable dependence for MTS forecasting.

**The effect of polynomial order.** We further investigate the relation between approximation error and the polynomial order $K$, where the $K$ ranges from 1 to 4, and $K = 1$ corresponds to a linear approximation. The first two charts of Figure 3 (b) present the average MFE and corresponding std, respectively. The results show that the approximation variance increases significantly with $K$, especially when $K = 4$. Moreover, TPGNN-K4 has a large MFE on the six datasets compared with other configurations. Therefore, we think a high-order matrix polynomial impairs both the precision and robustness of the dependence capturing ability of TPGNN. We point out that the observation is non-trivial since GPR-GNN achieves good results with a large $K$ ($K \geq 10$) on normal graph datasets, possibly due to the difference between MTS forecasting and node classification. The third chart shows the prediction error, it also validates that model with low MFE has a low prediction error.

**Case study on the real-world dataset.** Due to the absence of ground-truth dependence, we illustrate the correlation modeling ability of TPGNN on real-world datasets with an example of PEMS-D7. We find three highly correlated nodes in the learned result but independent in the prior structure, where the node IDs are 60, 93, and 218. In Figure 4 (a), we mark the physical location of the three nodes according to PEMS [3]. The three nodes are on the same road, which illustrates that TPGNN can learn reliable dependence from data. In Figure 4 (b), we show the learned correlation of pairs $(60, 218)$ and $(93, 218)$. We observe that $(93, 218)$ has a stable dependence and $(60, 218)$'s correlation increases in the evening, the difference possibly caused by the branch road between 218 and 60.

# 6 Conclusion

We proposes a novel GNNs-based method for MTS forecasting, TPGNN, which represents the variable dependence as a temporal polynomial matrix. The results on six real-world datasets illustrate the outstanding forecasting performance of TPGNN. Besides, extensive experiments on synthetic datasets demonstrate that our method achieves the best dependence approximation than other baselines.

**Limitations and future works**    Although TPGNN achieves outstanding performance on MTS forecasting and dependence capturing, our methods have limitations. First, the dependence captured by TPGNN is not a strict causal structure. Therefore, some correlations are unreliable and decrease the model's robustness. Second, MTS forecasting in the real world is affected by rare events like weather disasters, which may significantly impair prediction accuracy. The future works focus on causal learning and transfer learning [9, 24, 28, 38] on MTS data to solve the above limitations.

# 7 Acknowledgment

The work of the authors was supported by the National Natural Science Foundation of China (No. 62132017).

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
