# OpenReview forum: "Multivariate Time-Series Forecasting with Temporal Polynomial Graph Neural Networks"
_NeurIPS.cc/2022/Conference — NeurIPS 2022 Accept_

### Official Review · Reviewer_gX4A · 2022-07-06

**Rating:** 5
**Confidence:** 5
**Soundness:** 2 fair
**Presentation:** 3 good
**Contribution:** 2 fair

**Summary:**

This paper focuses on multivariate time series forecasting based on correlation graphs between such data. The main motivation is that dynamic correlation can avoid the bias in the solution that approaches focusing on the static correlation graph operate. Leveraging this dynamic correlation graph, a GNN-based model is leveraged to find higher-order representations of the data. One key aspect of the paper is that is uses time-varying coefficients (later, I argue that it is not explicitly the case). Some theoretical comparisons on the graph-based component of the paper are given. Numerical result on several datasets show a good performance and a thorough abolition study supports the role of the different components.

**Questions:**

1- All work is based on correlation graphs to support forecasting. But in real multivariate time series, causation is often more important than correlation, especially in traffic networks. And since the correlation does not imply causation, the model presents a limitation in this regard. The authors should discuss this aspect int he paper and provide guidelines about it.

2- Regarding the challenges of related works in the introduction. Point 1) about no variable dependence and using physical distance graphs. This is one approach to build the graph as the physical graph may provide additional information to aid the forecasting. But those methods could apply to any graph, even a similarity (KNN graph, correlation) and the likes. However, working with correlation graphs (or in general data-graphs) may not necessarily be beneficial. This is because both the graph and the model parameters are estimated from the same data, hence there is redundancy in the model. In this regard, the graph structure does not bring any additional information that is not present in the data themselves (as a road network would do) but it rather acts as bias (the type of graph) we choose to model the data. In this regard, the scientific challenge of current approaches are not well positioned.

3- Following the above point, the fact that temporal data may be better represented by time-varying graphs is of more importance. The paper should however acknowledge and contrast the contribution also with alternatives coming from non-GNN but graph-based methods such as [R1, R2, R3] and references therein. Also there are also works on GNNs over dynamic graphs, e.g., [R4] and following works that refer to it. The same remark applies here as well.

4-Wrt works [36, 35, 4] it is claimed that those models also work with a dynamic correlation among variables. And it is also claimed that their issues is that the learned graph is sensitive to the observation. Can you elaborate more on this, why it is the case. And also why in the proposed model does not suffer from it as the graph is also build from the observation. In my view, this is needed as the proposed approach is related to these works and this may entail the key contribution.
	4a) In general, it I given the impression in the introduction that one of the key novelties of this paper is to work with time varying graphs. But as acknowledged here and in comment 2 other works have already proposed this. In this regard, the paper introduction does not give merit to existing literature and does not detail entirely the key contribution of the paper wrt those alternatives.

5- Related works on GNNs and PGFs.
	5a) The division of GNNs into spectral and spatial methods is misleading. While there is a track of record within the CS community to do so, recent works have indeed realised that all GNN approaches are spatial—maybe the only spectral one is the early work in [R5]. In fact, the GCN mentioned as a spectral method is actually spatial. It can be seen as a particular case of GraphSage or message passing where the aggregation is dictated by multiplying nodal features with the graph Laplacian.
	5b) The GCN is a particular case of [R6] that claimed to use Chebyshev polynomials but as realised recently they often learn coefficients not respecting the coefficient rules in the Chebyshev polynomials [R8]. In turn, since the GCN is just an order one polynomial, and a particular case of [R6], it is more appropriate to give as an example [R6].
	5c) Polynomial graph filters have been initially formalised in the field of graph signal processing [R9, R10]. Also GCN [12] and ChebNet [R6, R8] use polynomial graph filters in the form of Chebyshev polynomials [R11]. The work in [30] instead is a very particular case of polynomial filtering. The link between polynomial filtering and GNNs has been discussed in [R12] including the expression of GCNs and SGCs as such polynomials. Another broader category casting the aforementioned GNNs are particular instances of graph filtering can be found in [R13] and their use for spatiotemporal learning has been discussed in [R14, R15, R16].
	5d) Some encoder decoder structure for GNN-based learning in multivariate time series have been discussed in [R17, R18]. The paper should highlight the contribution wrt these works as they also focus on dynamic graphs within the encoder-decoder solution.

While some of this discussion may not be central to the developed method, I feel it is propagating misleading information about the GNN architecture. The authors are in this regard requested to reorganise this part and highlight the differences in using PGFs wrt to alternatives that have used them for multi-variate time series forecasting.

6- Method

6a) Learning matrices E in (2) has a number of parameters Nc. Isn’t this too much and suffering the curse of dimensionality?

6b) The graph filter with time-varying coefficients has been discussed for forecasting in [R14, R16]. Here the difference stems mainly on using a (partially) learnable matrix A as well. Please highlight the differences.

6c) After (5): the use of “average result” yields a smoothing (or low pass filtering) of the spatiotemporal data; hence, while it can play a role in robustness it also affects performance. Limitations of this choice should be discussed.

6d) From eq. (7), it is shown that the average coefficients are used. So the final model in the end is not strictly a time varying model. This comes as a surprise as up to this point is has been build up towards a method that was time varying. It is therefore needed to highlight this properly not only in the introduction but also when contrasting with related works. The model is basically some engineered form of a PGF applied per time stamp, where the coefficients are the same (they only differ in the embedding). This to some extent contradicts the motivation and the theme of the paper in the earlier parts, which should be re-organised to make this part explicit.

6e) Why the decoder is not made graph-based? Can one then attribute the benefits to the graph-based solution proposed in the paper or to this non graph-based decoder? The motivation behind such a choice and role of the different components is unclear.

7- Theoretical properties - Thm1 is not about the TPGNN but it is rather the ability of a graph filter wrt the adjacency matrix A to approximate a user-defined operator. Similar results have been already discussed in the field of graph signal processing; see e.g., [R9, R19]. This result instead is applied to every operator (G^t) and errors are summed. The polynomial order in fact does not have to be large enough, but should satisfy the Cayley-Hamilton theorem (see [R19]).

While we could argue this result differs so some extend form that in the aforementioned works, I feel it is tangential to the method (TPGNN) as it does not account for several aspects of the method, e.g., (7) and the diffusion convolutional layer where these are embedded. Consequently, it is unclear if the TPGNN can indeed achieve perfect approximation and the provided theoretical properties reported do not support it.

8- In Sec. 5.5. the definition of the normal distribution, \sigma be a covariance matrix. Also it is said that the matrix signal X^(t) but the dimensions are those of a vector. I’m a bit confused here. In the same sentence it is also said that X^(t) depends on W^(t) but in the normal distribution it appears W^{(t-1)}. Are all these typos? If not considering the entries of X^(t) independent  and identically distributed seems like a bit limitation to me. Can the authors elaborate why it makes sense considering this way of generating the data?

Minor comments
- Sec. 3.1: The MTS data is a graph signal \mathcal{G}… is a wrong statement as the symbol represents a set (i.e., the graph), while the graph signal is defined as the data on this graph [R9-R10]-R14-R16-R11.
- Sec. 3.1.-If W^t is time varying, shouldn’t also \mathcal{E} be a time varying set of edges? Or do you mean a set of fixed edges exist and only their weights changes?
- “not independent“ -> dependent
- Is any link between the embedding vectors e_ts in the paragraph after (4) and the embedding matrix E in (3)? Either way, please make this explicit.
- In 3.2, Laplacians are indicated with bold L; in Thm.1 with bold G;
- There is a typo in the body of Lemma 2: “mathbfG”
- Are there enough data in the test set to have a significant fourth decimal digit (e.g., Tab2)


[R1] Natali, Alberto, et al. "Learning Time-Varying Graphs from Online Data." arXiv preprint arXiv:2110.11017 (2021).

[R2] Shafipour, Rasoul, et al. "Identifying the topology of undirected networks from diffused non-stationary graph signals." IEEE Open Journal of Signal Processing 2 (2021): 171-189.

[R3] 	.	B.Zaman,L.M.L.Ramos,D.Romero,andB.Beferull-Lozano,“Online  topology identification from vector autoregressive time series,” IEEE Transactions on Signal Processing, vol. 69, pp. 210–225, 2021.

[R4] Rossi, E., Chamberlain, B., Frasca, F., Eynard, D., Monti, F., & Bronstein, M. (2020). Temporal graph networks for deep learning on dynamic graphs. arXiv preprint arXiv:2006.10637.

[R5] Bruna, J., Zaremba, W., Szlam, A., & LeCun, Y. (2013). Spectral networks and locally connected networks on graphs. arXiv preprint arXiv:1312.6203.

[R6] Defferrard, Michaël, Xavier Bresson, and Pierre Vandergheynst. "Convolutional neural networks on graphs with fast localized spectral filtering." Advances in neural information processing systems 29 (2016).

[R8] He, Mingguo, Zhewei Wei, and Ji-Rong Wen. "Convolutional Neural Networks on Graphs with Chebyshev Approximation, Revisited." arXiv preprint arXiv:2202.03580 (2022).

[R9] Sandryhaila, Aliaksei, and José MF Moura. "Discrete signal processing on graphs." IEEE transactions on signal processing 61.7 (2013): 1644-1656.

[R10] Shuman, David I., et al. "The emerging field of signal processing on graphs: Extending high-dimensional data analysis to networks and other irregular domains." IEEE signal processing magazine 30.3 (2013): 83-98.

[R11] Shuman, David I., et al. "Distributed signal processing via Chebyshev polynomial approximation." IEEE Transactions on Signal and Information Processing over Networks 4.4 (2018): 736-751.

[R12] Gama, Fernando, et al. "Graphs, convolutions, and neural networks: From graph filters to graph neural networks." IEEE Signal Processing Magazine 37.6 (2020): 128-138.

[R13] Isufi, Elvin, Fernando Gama, and Alejandro Ribeiro. "EdgeNets: Edge varying graph neural networks." IEEE Transactions on Pattern Analysis and Machine Intelligence (2021).

[R14] Isufi, Elvin, and Gabriele Mazzola. "Graph-time convolutional neural networks." 2021 IEEE Data Science and Learning Workshop (DSLW). IEEE, 2021.

[R15] Hadou, Samar, Charilaos I. Kanatsoulis, and Alejandro Ribeiro. "Space-time graph neural networks." arXiv preprint arXiv:2110.02880 (2021).

[R16] Isufi, Elvin, et al. "Forecasting time series with varma recursions on graphs." IEEE Transactions on Signal Processing 67.18 (2019): 4870-4885.

[R17] Sanchez-Gonzalez, Alvaro, et al. "Learning to simulate complex physics with graph networks." International Conference on Machine Learning. PMLR, 2020.

[R18] Pfaff, Tobias, et al. "Learning mesh-based simulation with graph networks." arXiv preprint arXiv:2010.03409 (2020).

[R19] Segarra, Santiago, Antonio G. Marques, and Alejandro Ribeiro. "Optimal graph-filter design and applications to distributed linear network operators." IEEE Transactions on Signal Processing 65.15 (2017): 4117-4131.

**Limitations:**

see above in the questions

**Strengths And Weaknesses:**

Strengths:
+ Works on dynamic graphs for forecasting rather than static, which is a less exposed area;
+ Establishes links between;
+ Numerical results are quite strong and support the method;
+ Some theoretical results about the role of the graph filter (not of all the method) are given;

Weaknesses
- The paper over claims its contribution as proposing learning solutions over time varying graphs; while later on discusses that there exist prior work on learning over time-varying graphs;
- Related work are rather limited and discusses mainly some baseline methods in GNN but misses several recent works on polynomial filtering for MTS; hence, the specific contribution of the paper is unclear and not well positioned, consequently, its novelty is lower than claimed.
- The paper focuses entirely on correlation as the main argument to aid forecasting, bypassing any discussion related to causation; hence, limitations of the approach are not thoroughly discussed;
- The provided theoretical result are tangential to the method developed and rather focused on the graph filter but not of the overall solution;
- It is unclear if the benefits comes from the graph-based encoder with all the dynamics or from the non-graph based decoder.

---

> ### Author Response · Authors · 2022-08-02
> **We updated the related works and discussed the limitations according to your advice.**
>
> Thank you, your comments are valuable and helpful for revising and improving our paper and providing necessary guidance for our research.
>
> Q1-2. A:  Although we do not explicitly capture the causation structure, the regularization terms to the polynomial coefficients and matrix basis help TPGNN to eliminate redundant dependence. According to the experimental results, the learned structure is highly sparse, with a Frobenius norm of $16.12 \pm 1.31$ (the matrix size is $288\times 288$). Besides, if we replace the learned structure with the prior structure, i.e., the physical distance graph, then the performance declines by $13.13$%, $13.94$%, and $12.23$% on the three metrics. We discussed the limitations in Section 6.
>
> Q3 & Q5. A: We updated the related works for discussing the research you mentioned. Besides, we merged the discussion on GCNs and PGFs into one paragraph.
>
> Q4. A: These works model dynamic correlation among variables based on self-attention. For self-attention, the input sequence decides the constructed structure. However, the input length on the PEMSD7 dataset is 12, but the whole training sequence has a size of over 8000. Therefore, the dynamic structure captured by self-attention is sometimes misleading, especially when the input sequence is noisy. Besides, the main motivation of these works for using self-attention is providing input for the GNNs. In our work, we explicitly point out the challenge and propose a practical method that accurately models dynamic dependence.
>
> Q6. A: a) N and c are small in practice. In Appendix 4, we list the parameter scale of different methods. As a snapshot, TPGNN: 0.31M, MTGNN: 0.44M, Graph Wavenet: 0.25M, STGCN: 0.33M. Therefore, our method TPGNN is lightweight in parameter scale.
>
> b) According to Eq (7) of [R14] and Eq (13) of [R16]. The two works do not actually adapt the coefficients according to the time, they share a set of polynomial coefficients across different sliding windows.
>
> c) In Table 3 of the ablation study, we investigate the influence of the average coefficients on the model performance. w/o overview is the version that does not use average coefficients. The results indicate that TPGNN outperforms the w/o overview's accuracy by 3.60% on average, and the w/o overview has a significant accuracy variance. Besides, predicting with $T$ groups of polynomial coefficients increases the space complexity.
>
> d) According the Equation (5) and (6), we have $\mathbf{\bar{a}}=(\mathbf{e}^{(t)},\dots,\mathbf{e}^{(t+T-1)})\mathbf{W}_c\mathbf{W}_a$. Clearly, $\mathbf{\bar{a}}$ is decided by $t$, i.e., the average coefficients are still time-varying. $\mathbf{\bar{a}}$ changes with the movement of the sliding window according to the equation.
>
> e) We further implement a graph-based-decoder version of our method, the experimental results on PEMSD7 (horizon 12) are (MAE/MAPE/RMSE): $3.349\pm0.137/8.461\pm0.133/6.680\pm0.330$. The change impairs the performance by 2.92%. Due to the auto-regressive process, we have to train a recurrent GNN if we use a graph-based decoder. The over-smoothing problem of GNN is unavoidable because we share an identical GNN across the iterations.
>
> Q7:  Theorem 1 is used to address the 2nd challenge. It points out the ability boundaries of the TPG module, which gives the requirements to achieve perfect approximation. Otherwise, the approximation ability of our method is unknown for general cases. We have carefully checked R[9] and R[19], but they do not have a result like Theorem 1 to specify the bound of the TPG module under the Frobenius norm. We further generate synthetic MTS datasets with ground-truth variable dependence to solve the second challenge. Theorem 1 and the synthetic datasets experiments are indispensable to validate the TPGNN's effectiveness.
>
> Q8: At time step $t-1$, we define a temporal graph structure $\mathbf{W}^{(t-1)}\in\mathbb{R}^{N\times N}$as the ground-truth dependence. We then aggregate $\mathbf{X}^{(t-1)}\in\mathbb{R}^{N\times 1}$ based on $\mathbf{W}^{(t-1)}\in\mathbb{R}^{N\times N}$, i.e., $\mathbf{W}^{(t-1)}\mathbf{X}^{(t-1)}$. We set $\mathbf{W}^{(t-1)}\mathbf{X}^{(t-1)}\in\mathbb{R}^{N\times 1}$ as the mean, $\sigma$ as the standard deviation, and produce  $\mathbf{X}^{(t)}$by sampling from the normal distribution $\mathcal{N}(\mathbf{W}^{(t-1)}\mathbf{X}^{(t-1)},\sigma)$. The detailed algorithm is shown in Appendix A.3.3. There are two reasons to use the method. 1) The method enables us to access the ground-truth variable dependence. 2) The method is easy to implement, and the resultant MTS data has complex behavior. We have visualized some examples in Appendix A.3.3.
>
> Minor comments.1-3) A: Thanks for your suggestions. We update Section 3.1 to clarify some misleading concepts. (4) They have no connections. 5) the latter represents the ground-truth laplacians. 6) Solar-energy has 10512 test samples, the sample numbers are listed in Table1.

---

> ### Author Response · Authors · 2022-08-09
> **Looking forward to further comments!**
>
> Dear Reviewer,
>
> We have updated our related works and further discussed the limitations. We are wondering if our response and revision have cleared your concerns. We would appreciate it if you could kindly let us know whether you have any other questions. We are looking forward to comments that can further improve our current manuscript. Thanks!
>
> Best regards,
>
> The Authors

---

> > ### Author Response · Authors · 2022-08-10
> > **Looking forward to further discussion!**
> >
> > Dear reviewer,
> >     Do you have any further concerns or suggestions? We are very delighted to discuss them with you.

---

### Official Review · Reviewer_3dhS · 2022-07-10

**Rating:** 5
**Confidence:** 4
**Soundness:** 3 good
**Presentation:** 4 excellent
**Contribution:** 3 good

**Summary:**

This paper introduces a Temporal Polynomial Graph Neural Network (TPGNN) for multivariate time series forecasting. TPGNN first models the overall correlation with a static matrix basis, and then constructs a matrix polynomial for each time step by a set of learnable time-varying coefficients and the matrix basis. The experimental results on several benchmark datasets show that TPGNN could outperform the baseline methods.

**Questions:**

1. Self-attention might be the simplest way to learn dynamic graph structures. How does it perform compared to TPGNN?
2. What's the time complexity of TPGNN?


**Strengths And Weaknesses:**

Strengths:
1. The experimental results on PEMS datasets show that the proposed TPGNN could achieve state-of-the-art performance.
2. The experimental results on the synthetic dataset show that TPGNN could approximate the underlying graph structure.
3. The paper is clear and easy to follow.


Weakness:
1. The novelty of using matrix polynomial to approximate graph structure is limited, as it is quite similar to [1].
2. The experimental results on the single future step forecasting task show that the improvements are not significant for many metrics.
3. Other weaknesses: please refer to the questions.

[1] Chien, Eli, et al. "Adaptive universal generalized pagerank graph neural network." arXiv preprint arXiv:2006.07988 (2020).

---

> ### Author Response · Authors · 2022-08-02
> **Your concerns about GPR-GNN [1] and self-attention have been discussed in our paper.**
>
> Thank you, your comments are valuable and helpful for revising and improving our paper, as well as providing important guidance for our research.
>
> We have noticed that our work has similarities to GPR-GNN [1]. In the related work, we thoroughly discuss the difference between GPR-GNN and our work. The main difference is that our method can capture variable dependence of multivariate time-series (MTS) data by representing the dependence as a temporal matrix polynomial. It is crucial for the GNNs-based MTS forecasting method since MTS data commonly do not have an explicit graph structure, which is different from typical graph datasets like Cora and DBLP.
>
> Furthermore, we generate six synthetic MTS datasets that have ground-truth variable dependence. We then compare the dependence capturing ability of several baselines with our method. The results demonstrate that our approach outperforms GPR-GNN by 53.47% in approximating the dependence structure.
>
> Q: Self-attention might be the simplest way to learn dynamic graph structures. How does it perform compared to TPGNN?
>
> A: We have compared our method with self-attention in Table 4,  the results demonstrate that our method outperforms self-attention by 23.41% in capturing the variable dependence. Furthermore, we implement a self-attention version of our method and test it on the PeMSD7 dataset. The results for horizon 12 are (MAE/MAPE/RMSE): $4.325\pm 0.063/10.553\pm 0.204/8.040\pm0.146$, which impairs the accuracy by 28.42% on average. As a result, self-attention is impractical for capturing the dynamic graph structures.
>
> Q: What's the time complexity of TPGNN?
>
> A: Suppose that the input MTS data has $N$ variables, the input sequence length is $T$, the feature dimension is $D_e$, and the polynomial order is $K$. The TPGNN is composed of self-attention and the TPG module, and it is known that self-attention has a time complexity as follows:
> $$
> \mathcal{O}(NT^2D_e+NTD_e^2)
> $$
>
> As for the TPG module, the time complexity for constructing the initial adjacency matrix is $\mathcal{O}(N^2)$, and the time complexity of Equation (7) is $\mathcal{O}(KT(ND_e^2+N^2D_e))$. As a result, The overall time complexity of TPGNN is $\mathcal{O}(KTN^2D_e+KTND_e^2+NT^2D_e)$. Since the $K, D_E$ is small in practice, the efficiency of TPGNN is mainly decided by $N$ and $T$. We add a section in Appendix A.2 to comprehensively illustrate the time complexity of our method. Besides, Appendix A.2 also contains numerical results of TPGNN's efficiency. The results show that TPGNN has competitive efficiency with SOTA methods like MTGNN [39]/Graph Wavenet [40], though these methods do not have self-attention modules.

---

> > ### Comment · Reviewer_3dhS · 2022-08-09
> > **Response**
> >
> > Thanks for the response. My concerns are addressed.

---

### Official Review · Reviewer_U7ug · 2022-07-13

**Rating:** 7
**Confidence:** 4
**Soundness:** 3 good
**Presentation:** 4 excellent
**Contribution:** 3 good

**Summary:**

The paper presents a novel framework for multivariate time series forecasting in the context of Graph Neural Networks (GNN). The aim of the work is to take into account the variable dependence which can evolve temporally.   To tackle the issue, the authors propose to use a dynamic adjacency matrix inside the GNN process. This matrix corresponds to a polynom of an initial adjacency matrix where the coefficients are learned from the data and depend on the timestamp.  The initial adjacency matrix is computed from the data following previous works and can integrate previous knowledge. A hypothesis of periodicity is considered in order to obtain a finite number of polynomial coefficients. The computed dynamic adjacency matrix is then used in a Graph Convolutional network to obtain the encoding module. The decoder uses a self-attention mechanism to forecast in a auto-regressive way the future of the time series. The authors propose a theorem to prove the correctness of the estimation of the adjacency matrix.  An extensive experimental part compares the proposed approach to several SOTA baselines and presents a complete ablation study. Finally, experiments on artificial (and real) data show the effectiveness of the proposed module to learn the graph correlation structure.



**Questions:**

From what I understand, the initial adjacency matrix is computed from embeddings that are not related to the learning process. It is correct? Do think it is possible to learn it in an end-to-end way?

**Strengths And Weaknesses:**

Strengths:
* The paper is well written, the SOTA is clearly stated and the work is well positioned wrt to previous works.
* The model is clearly presented and illustrated
* The experimental part is very well designed, with exhaustive experiments (comparison to SOTA models, ablation study wrt all the aspects of the model,  analysis of the captured correlation,..).

Weaknesses :
* The novelty is not great (but sufficient in my opinion)
* The experimental section is quite overfilled, maybe some results can be put to the appendix to give more space to introduce the datasets for instances.

---

> ### Author Response · Authors · 2022-08-02
> **The initial adjacency matrix is learned in an end-to-end way.**
>
> Thank you, your comments are valuable and helpful for revising and improving our paper, as well as providing important guidance for our research.
>
> Q: From what I understand, the initial adjacency matrix is computed from embeddings that are not related to the learning process. It is correct? Do think it is possible to learn it in an end-to-end way?
>
> A: In the equation(2) and (3), we construct the initial adjacency matrix with embeddings $\mathbf{E}$：
> $$
> \begin{equation}
> \mathbf{A}=\operatorname{SoftMax}(\operatorname{ReLU}(\mathbf{EE^T}))
> \end{equation}\tag{2}
> $$
> $$
> \begin{equation}
> \mathbf{A}=\operatorname{SoftMax}(\operatorname{ReLU}(\mathbf{EE^T}))+\mathbf{L}
> \end{equation}\tag{3}
> $$
>
> Following the end-to-end fashion, these embeddings are also learnable parameters. As a result, we optimize the resultant adjacency matrix in the learning process (end-to-end).

---

> > ### Comment · Reviewer_U7ug · 2022-08-08
> > **Rebuttal feedback**
> >
> > Ok, thank you for the  clarifications

---

### Official Review · Reviewer_ppp4 · 2022-07-14

**Rating:** 5
**Confidence:** 5
**Soundness:** 2 fair
**Presentation:** 3 good
**Contribution:** 3 good

**Summary:**

The paper proposes TPGNN, a novel graph model for multivariate time-series forecasting. The most significant improvement of TPGNN is the ability to learn dynamic correlations between nodes. TPGNN models a dynamic graph with a novel component, TPG, based on matrix polynomials. The paper demonstrates how the proposed model can learn actual correlations in a simulated setting and provide theoretical guarantees for the approximation. TPGNN achieves SoTA performance on 6 benchmark datasets, on both single-step and multi-step settings.

**Questions:**

- Many recent studies (such as Informer, Autoformer, FEDFormer, and N-HiTS, among others) use the Electricity, Traffic, and Exchange datasets for the long-horizon multi-step forecasting setting. Why didn't the authors consider these datasets as benchmarks for this task?
- How does the choice of T_p affects the performance? Would it be beneficial to match the cycle to known seasonalities of the data?
- Adding simpler univariate baselines to the experiments (classic models such as ARIMA, SeasonalNaive, and DL methods such as N-BEATS) will help better measure the complexity/accuracy tradeoff of TPGNN.

**Limitations:**

Limitations are discussed, and I do not identify potential negative social impacts.

**Strengths And Weaknesses:**

Strengths:
- The paper proposes a novel module, TPG, to model dynamic correlations between time series.
- Authors provide theoretical approximation guarantees and empirical results on simulated data.
- Ablation studies demonstrate the gains in performance of each component.
- TPGNN achieved SoTA performance on 6 benchmark datasets, on both single-step and multi-step settings.
- The paper is well written and clear.

Weaknesses:
- The model can only learn periodic correlations, as the coefficients of the polynomials are fixed between cycles. This potentially limits the model's advantages to settings with distinct periodic patterns. This might explain why TPGNN performs worst in Exchange, a dataset without clear seasonalities.
- To produce multi-step forecasts, TPGNN uses an auto-regressive strategy. This strategy performance is worst than other methods (such as the direct strategy) on long-horizon forecasting, as 1. errors accumulate over time, leading to worse performance, and 2. longer inference times.
- The paper claims the model achieves SoTA performance on the long-horizon setting; however, the horizons considered in the paper are far lower than most recent studies. For example, the Informer paper (and a whole body of literature) considers a multi-step forecast of 720 timestamps.
- The paper does not present any time complexity analysis.

---

> ### Author Response · Authors · 2022-08-02
> **We updated the baselines and added experiments according to your advice**
>
> Thank you, your comments are valuable and helpful for revising and improving our paper, as well as providing important guidance for our research. As for your concerns about the autoregressive (AR) strategy, the AR strategy has been proven effective across various areas. The strategy is still applied in frontier research like Denoising Diffusion Probabilistic Model (DDPM) [R1], and the most famous generation models like DALLE 2 [R2] and Imagen [R3] are based on DDPM. Although the prediction errors are accumulated over time, the strategy provides a powerful inductive bias to enforce the model to utilize the context information, improving the long-term prediction performance. As for the efficiency issue, we can reduce the computation cost by lowering the sample rate of data, e.g., increasing the time interval between two successive forecasts. Furthermore, we add a comprehensive time complexity analysis of TPGNN in Appendix 2.
>
> Q: Many recent studies (such as Informer, Autoformer, FEDFormer, and N-HiTS, among others) use the Electricity, Traffic, and Exchange datasets for the long-horizon multi-step forecasting setting. Why didn't the authors consider these datasets as benchmarks for this task?
>
> A: Thank you for your suggestion. The benchmark selections are mainly based on related works like MTGNN [39] and StemGNN[4]. They use the Electricity, Traffic, and Exchange datasets for evaluating model performance on the single-step forecasting. For the multi-step forecasting task, the evaluation is conducted on the traffic datasets. The two traffic datasets are essential for our evaluation since they have prior structures, i.e., the physical distance among sensors. The experimental results illustrate that the learned structure captures valid dependence that does not record by the prior structure. Furthermore, we compare our method with Informer [46] on the two traffic datasets. The results validate that the complexity of the two datasets and TPGNN outperforms other baselines significantly.
>
> Q: How does the choice of T_p affect the performance? Would it be beneficial to match the cycle to known seasonalities of the data?
>
> A: We commonly set $T_p$ as one day/week/season according to the sample rate of datasets. To answer your question, we conduct several experiments on PeMSD7 to investigate the influence of $T_p$. We set the $T_p$ as 16hour, 24hour (one day), 32hour, and the horizon is 12.
>
> The results for $Tp=16hour$ are (MAE/MAPE/RMSE): $3.241\pm0.021/8.248\pm0.089/6.547\pm0.016$
>
> The results for $Tp=24hour$ are (MAE/MAPE/RMSE): $3.215\pm0.014/8.221\pm0.035/6.559\pm0.023$
>
> The results for $Tp=32hour$ are (MAE/MAPE/RMSE): $3.234\pm0.024/8.230\pm0.077/6.536\pm0.052$
>
> The results illustrate that our method is robust to the $T_p$'s selection. Therefore, a roughly correct selection is enough to get a good result. As for the inferior performance on Exchange, we think the main reason is the small sample size, which causes difficulties in capturing the dynamic variable dependence.
>
> Q: Adding simpler univariate baselines to the experiments (classic models such as ARIMA, SeasonalNaive, and DL methods such as N-BEATS) will help better measure the complexity/accuracy tradeoff of TPGNN.
>
> A: Thanks for your suggestion. VARMLP [43] is one of the representative univariate baselines based on ARIMA. For the multi-step forecasting task, the FC-LSTM is a univariate baseline; Furthermore, we add ARIMA to the baselines according to your advice (Table 2).
>
> [R1] Nichol, Alex and Prafulla Dhariwal. “Improved Denoising Diffusion Probabilistic Models.” ArXiv abs/2102.09672 (2021): n. pag.
> [R2] Ramesh, Aditya et al. “Hierarchical Text-Conditional Image Generation with CLIP Latents.” ArXiv abs/2204.06125 (2022): n. pag.
> [R3] Saharia, Chitwan et al. “Photorealistic Text-to-Image Diffusion Models with Deep Language Understanding.” ArXiv abs/2205.11487 (2022): n. pag.

---

> ### Author Response · Authors · 2022-08-10
> **Looking forward to further comments!**
>
> Dear Reviewer,
>
> Since the rebuttal discussion is about to end soon, we are wondering if our response and revision have cleared your concerns. We would appreciate it if you could kindly let us know whether you have any other questions. We are looking forward to comments that can further improve our current manuscript. Thanks!
>
> Best regards,
>
> The Authors

---

### Author Response · Authors · 2022-08-07
**Looking forward to further discussions!**

Dear Reviewers,

We were wondering if our response and revision have cleared all your concerns. In the previous responses, we have tried to address all the points you have raised. In the remaining 3 days of the rebuttal period, we would appreciate it if you could re-evaluate our submission, or kindly let us know whether you have any other questions, so that we can still have time to respond and address them. We are looking forward to discussions that can further improve our current manuscript. Thanks!

Best regards,

The Authors

---

### Author Response · Authors · 2022-08-09
**Looking forward to further discussions!**

Dear Reviewers,

We were wondering if our response and revision have cleared all your concerns. In the previous responses, we have tried to address all the points you have raised. In the remaining 1 day of the rebuttal period, we would appreciate it if you could re-evaluate our submission, or kindly let us know whether you have any other questions, so that we can still have time to respond and address them. We are looking forward to discussions that can further improve our current manuscript. Thanks!

Best regards,

The Authors

---

### Meta-Review · Area_Chair_Fczz · 2022-08-30

**Recommendation:** Accept
**Confidence:** Certain

**Metareview:**

This well-written paper has been carefully evaluated by four competent reviewers. Three of them rated the work as marginally acceptable, one gave it full accept score. In despite of a few identified deficiencies, including limited cohort of comparison models, overstated claims about performance of the proposed model at long-range forecasting, and some minor limitations of the empirical evaluation protocol, the reviewers were confidently positive about the work. I recommend acceptance.

**Award:**

No

---

### Decision · Program_Chairs · 2022-09-14

Accept